# Inflammation-Responsive Nanovalves of Polymer-Conjugated Dextran on a Hole Array of Silicon Substrate for Controlled Antibiotic Release

**DOI:** 10.3390/polym14173611

**Published:** 2022-09-01

**Authors:** Ai-Wei Lee, Pao-Lung Chang, Shien-Kuei Liaw, Chien-Hsing Lu, Jem-Kun Chen

**Affiliations:** 1Department of Anatomy and Cell Biology, School of Medicine, College of Medicine, Taipei Medical University, Taipei 106, Taiwan; 2Department of Materials and Science Engineering, National Taiwan University of Science and Technology, Taipei 10607, Taiwan; 3Department of Electronic and Computer Engineering, National Taiwan University of Science and Technology, Taipei 10607, Taiwan; 4Department of Obstetrics and Gynecology, Taichung Veterans General Hospital, Taichung 40705, Taiwan; 5Ph.D. Program in Translational Medicine, Institute of Biomedical Sciences, Rong-Hsing Research Center for Translational Medicine, National Chung-Hsing University, Taichung 40227, Taiwan

**Keywords:** PMAA brush, ROS, hole array, antibiotics, inflammation

## Abstract

Poly(methacrylic acid) (PMAA) brushes were tethered on a silicon surface possessing a 500-nm hole array via atom transfer radical polymerization after the modification of the halogen group. Dextran-biotin (DB) was sequentially immobilized on the PMAA chains to obtain a P(MAA-DB) brush surrounding the hole edges on the silicon surface. After loading antibiotics inside the holes, biphenyl-4,4′-diboronic acid (BDA) was used to cross-link the P(MAA-DB) chains through the formation of boronate esters to cap the hole and block the release of the antibiotics. The boronate esters were disassociated with reactive oxygen species (ROS) to open the holes and release the antibiotics, thus indicating a reversible association. The total amount of drug inside the chip was approximately 52.4 μg cm^−2^, which could be released at a rate of approximately 1.6 μg h^−1^ cm^−2^ at a ROS concentration of 10 nM. The P(MAA-DB) brush-modified chip was biocompatible without significant toxicity toward L929 cells during the antibiotic release. The inflammation-triggered antibiotic release system based on a subcutaneous implant chip not only exhibits excellent efficacy against bacteria but also excellent biocompatibility, recyclability, and sensitivity, which can be easily extended to other drug delivery systems for numerous biomedical applications without phagocytosis- and metabolism-related issues.

## 1. Introduction

Bacteria are currently one of the major causes of nosocomial infections, which may lead to many unattended infectious diseases that can progress to systemic inflammation and multiorgan dysfunction, threatening human beings in more severe cases [1]. These pathogenic bacteria include Gram-negative and Gram-positive bacteria responsible for many infections in human beings with severe and even fatal consequences [2]. Sepsis is a multiorgan dysfunction disease caused by bacteria that can present with a poor prognosis and potential long-term sequelae. Bacterial infection can induce the systemic inflammatory response syndrome of sepsis, which poses great threat to human life [3]. To improve the sepsis survival rates, preferable antimicrobial treatment is recommended within the first hour of suspecting sepsis [4]. However, antibiotic-resistant pathogens are still particularly problematic in the context of sepsis caused by bacterium infections, which cause poor outcomes. The inflammatory phase generally results in the increase of reactive oxygen species (ROS) production and a decrease in pH to ca. 5.7 because of the increase in metabolic activation [5,6]. Thus, pH and ROS are appropriate stimuli for materials targeting inflammation to improve therapeutic efficacy against antimicrobial resistance [7,8]. The ability of an automated and continuous system for administering antibiotics at precise doses and intervals in response to pH/ROS to avoid adverse effects such as the development of bacterial resistance predominately determines the therapeutic efficacy [9]. pH-responsive materials including biofilms [10], fibrous mats [11], and nanocarriers [12] have been employed to improve therapeutic efficacy against antimicrobial resistance. However, few ROS-responsive materials have been reported, especially for a pH/ROS dual-responsive drug delivery system with a container [13]. In addition, a system that can maintain its integrity during the antimicrobial treatment and allow for efficient drug release is also desirable.

Phenylboronic acid (PBA) is most commonly used as a ROS-sensing element through the cleavage of a C-B covalent bond that can consume ROS sensitively toward H_2_O_2_ [14,15,16]. In a previous study, the modification of carbohydrate dextran with PBAs into a ROS-responsive polymer was reported [17]. PBA can also be ionized upon binding to diols in response to pH/glucose, and it can be included as a pendant group in polymer chains [18]. Dextran, a kind of carbohydrate polymer, has been widely applied in biomedical fields due to its excellent biocompatibility, the low price of its commercial product, and its fluorescein diacetate (FDA)-approved properties [19]. Polymers binding with dextran through the hydroxyl groups have been reported in previous studies [20,21]. These dextran-based polymers can revert to their initial states in acidic solutions by hydrolyzing, which has been incorporated into pH-responsive hydrogels for release of drugs [22,23,24].

Smart polymer brushes represent the tethering of chain ends of stimulus-responsive polymer to a surface, resulting in the formation of different structures under various stimuli [25,26]. A polymer brush can anchor to several reactive sites arbitrarily during polymerization with various monomers, facilitating the immobilization of pendant groups [27,28]. Thus, they exhibit excellent flexibility for surface modification, which has been applied widely in bioassays [29] and cell adhesion [30], biomimetic substrate [31], antibacterial property [32], and solvent sensor [33] analyses. Various polymers responding to different stimuli, such as temperature, pH, light, ultrasound, electricity, and host–guest interactions, have also been developed [34]. Temperature and pH are the most common stimuli for constructing smart polymers based on their amphiphilic and ionizable characteristics, respectively [35]. Poly(methacrylic acid) (PMAA) possesses several carboxyl groups on its polymer chain, and it has been classified as a biocompatible polymer with thermo- and pH-responsive behaviors simultaneously [36,37]. COOH-rich PMAA chains can bind with amide groups through EDC/NHS (1-ethyl-3-(3-dimethylaminopropyl)carbodiimide/*N*-hydroxysuccinimide) activation, which provide an excellent conjugation strategy for biomacromolecule immobilization [38,39]. “Grafting to” and “grafting from” approaches are two approaches to tether the polymer grafts on the substrates through grafting polymerization [40,41]. In comparison with the grafting to approach, grafting from is more flexible for tailoring the polymer structure and density. Surface-initiated atom transfer radical polymerization (SI-ATRP) is the most common grafting from method due to its versatility and stability.

An ideal drug delivery system should combine sensitively stimuli-triggered valves and drug tanks to administer antibiotic treatment autonomously. To address these challenges, a regular hole array with a 500-nm resolution was fabricated by semiconductor process on silicon wafers to store the drug as nanotanks. Halogen groups were bound to the amide group-modified silicon surfaces and served as the initiator layer in ATRP. PMAA was grafted from the halogen groups on the silicon surfaces by SI-ATRP [42,43]. Dextran-biotin (DB) was employed to conjugate with COOH groups of the PMAA brush to obtain P(MAA-DB) brushes surrounding the holes. After loading with FITC-vancomycin inside each hole, the P(MAA-DB) brush was cross-linked by biphenyl-4,4′-diboronic acid (BDA) to form a boronate ester bond that can cap the holes and block the release of the drug (Figure 1). The cleavage of the bond formed by the binding between DB and BDA in the polymer-capped system is not stable under ROS conditions. The boronate esters disassociated in the presence of H_2_O_2_, leading to the opening of the holes and the release of the loaded drug. This strategy offers a convenient approach for an antibiotic delivery system with ROS-triggered valves that can be applied as an implanted chip for long-term antimicrobial treatment. 

## 2. Experimental Section

### 2.1. Materials

Six-inch polished silicon wafers (Hitachi, Tokyo, Japan) were employed to pattern a hole array on the substrate by VLSI. Each 25-mm square die possessing a 500-nm hole array was cut from the silicon wafer as the drug container. 3-aminopropyltriethoxysilane (3A; H_2_N(CH_2_)_3_Si(OC_2_H_5_)_3_; Mw: 221.37), 2-bromo-2-methylpropionyl bromide (2B; (CH_3_)_2_CBrCOBr; Mw: 229.90), and methacrylic acid (MAA; H_2_C=C(CH_3_)COOH; Mw: 86.09) for grafting polymers were obtained from Acros Organics Co (Geel, Belgium). Other components for SI-ATRP and EDC/NHS reactions were purchased from Sigma Aldrich (St. Louis, MO, USA) and utilized without further purification [44]. Biphenyl-4,4′-diboronic acid (BDA; Mw: 241.84, 94%,) was purchased from Thermo Fisher Scientific (Waltham, MA, USA). Dextran-biotin (CAT#: NTA-2011-ZP112; Mw: 10,000) was purchased from Creative Biolabs (Shirley, NY, USA). FITC labeled vancomycin (FITC-vancomycin, No: SBR00028, Mw: 1840.65) were also purchased from Sigma Aldrich.

### 2.2. Developing Antibiotic Delivery System with ROS-Triggered Valves

Figure 2 illustrates the construction of the antibiotic delivery system with ROS-triggered valves [45]. A and B: Photoresist was coated on silicon wafers to pattern hole array of 500-nm resolution by I-line lithography. C: Patterns of the samples were transferred to silicon wafer by inductive couple plasma etcher (Hakuto Corp, Tokyo, Japan) with SF_6_, and the residual photoresists were removed from the surfaces of silicon wafers, denoted as 500H.

D: After oxygen plasma treatment for 30 s to generate the hydrophilic groups on the surface, the samples were treated with 3A solution, and 2B solution was used to immobilize halogen groups on the surface as SI-ATRP initiators. A mixture of MAA, Cu(I)Br, CuBr_2_, and PMDETA in methanol/water (1:1), denoted as 500H-PMAA, was employed to synthesize PMAA brushes on the initiator-modified chip via ATRP following treatment for 8 h at 25 °C. DB units were conjugated to the carboxylic acid functional groups of PMAA brush as pendant groups by EDC/NHS reaction and denoted as 500H-P(MAA-DB) [46]. E: The prepared 500H-P(MAA-DB) was immersed in FITC-vancomycin solution and incubated for 24 h to load the drug inside the holes. F: The P(MAA-DB) brushes surrounding the hole edges, denoted as 500H-P(MAA-DB)-BDA [47], were cross-linked by BDA to cap the holes of the chip. A network of P(MAA-DB)-BDA was cleaved under a H_2_O_2_ concentration range of 5–20 mM to release the antibiotic from the holes. As-prepared samples could be loaded the antibiotic and the holes capped reversibly for recycling. Surface components and functional groups of the samples for each stage were analyzed by X-ray photoelectron spectroscopy (XPS; Scientific Theta Probe, Delta-T Devices, Cambridge, UK) and Fourier transform infrared spectrometry (FTIR, Digilab, FTS-1000, Holliston, MA, USA), respectively. The morphologies of the samples in dry and liquid states for each stage were observed using a field emission scanning electron microscope (FESEM; JEOL 7900F, Tokyo, Japan) and atomic force microscope (AFM; Veeco Dimension 5000 scanning probe microscope, Plainview, NY, USA) equipped with temperature control and liquid modules, respectively. 

### 2.3. Loading FITC-Vancomycin into the Chips and Its Release

The 25-mm square chip of 500H-P(MAA-DB) was immersed in 10 mL of FITC-vancomycin solution and incubated at 25 °C for 24 h, following which 50 mg of BDA was added to the solution to cap the holes and block the drug inside the holes by cross-linking the P(MAA-DB) brush for 3 h. The drug-loaded chips were removed from the solution and washed with phosphate-buffered saline (PBS) solution three times prior to use. 

The release of FITC-vancomycin from the holes of the chip was controlled with various H_2_O_2_ concentrations to develop an autonomous delivery system. A drug-loaded chip with polymer network cap was placed in 20 mL of PBS in the presence of H_2_O_2_ at various concentrations from 0 to 20 mM to evaluate the ROS-responsive behavior of drug release at a pH of 7.4. After the boronate ester bond of P(MAA-DB)-BDA network was dissociated by H_2_O_2_ within various functions of time to open the holes, 1 mL of PBS was withdrawn to obtain the fluorescence intensity of FITC-vancomycin at 515 nm. The release profile of drugs on the basis of concentration of FITC-vancomycin for long-term release control could be established. The chips were immersed in PBS solutions at pH 5.2, 6.1, 7.4 and 8.1 to evaluate the pH-responsive behavior of the drug release with a similar operation. After the dissociation of boronate ester bond in the presence of H_2_O_2_ at 10 nM to release FITC-vancomycin for 48 h, the chips were reloaded with the drug and retreated with BDA to cap the holes of chips for recycling experiments. Laser scanning confocal microscopy (LSCM; Leica TCS SP5 Confocal Spectral Microscope Imaging System, Leica Microsystems, Buffalo Grove, IL, USA) was utilized to observe the release of FITC-vancomycin from the holes.

### 2.4. Cell In Vitro Studies

The biocompatibility of the drug-loaded 500H-P(MAA-DB)-BDA chip was evaluated with the cell viabilities by seeding the L929 cells on the surfaces of chips. L929 cells were incubated in the medium supplemented with 10% fetal bovine serum, 100 U/mL penicillin, and 100 mg/mL streptomycin, refreshed ever two days. For the biocompatibility assays, the cells were seeded at 10^4^ cells/mL in each 96-well plate that filled with 150 μL medium without cells as blank references. 10^4^ cells were also seeded onto the sterilized chips with 150 μL medium for cell proliferation. The cell viabilities were evaluated by CCK-8 assay at 1, 3, 5 and 7 days.

A toll-like receptor 4 agonist, lipopolysaccharide (LPS), was employed to induce the production of ROS and evaluate the behavior of inflammation-triggered release in vitro by activating macrophages [48], and pH decreased in the microenvironment during the macrophage activation. Accordingly, the enhancement of macrophage activity leads to both pH and ROS stimuli, which can simulate an inflammatory microenvironment for evaluating the drug release and anti-inflammatory efficacy of the chips. 20,000 RAW 264.7 murine macrophages were grown and stimulated with LPSs at a concentration of 5 μg/mL for 2 h before incubation with the chips [48]. After various incubation times with chips, the media were collected to measure the released drug by fluorescence spectrometry at 515 nm using a plate reader (Molecular Devices SpectraMax M5, Sunnyvale, CA, USA). The macrophages without LPS activation after various incubation times were employed to evaluate the stability of the polymer network cap. The loading capacity of the chips, calculated by gravimetric analysis for FITC-vancomycin (52.4 μg/cm^2^) in the same medium was employed as a standard to normalize the collected media. In addition, fluorescence microscopy images of macrophages were also employed to verify the cellular uptake of drug. 

To trigger the inflammation-targeting efficacy of the antibiotics released from the chips, Raw 264.7 murine macrophages were activated by LPS (100 ng/mL). Each square-cut sample (0.5 cm × 0.5 cm) was incubated with 0.5 mL of LPS-activated macrophages solution, completely immersing the substrate for 2 h, to trigger the drug release by dissociating the cap of polymer network. After the dissociation of the cap, the chips were taken from the solution and then placed horizontally inside a 24-well plate and coincubated with Staphylococcus aureus (SA) in 0.5 mL of the medium, cation-adjusted Mueller Hinton Broth II (CMHB) per well. Briefly, 0.50 mL of SA in the exponential growth phase at 10^5^ CFU/mL in CMHB was injected in the well to submerge the chip. These 24-well plates including chip and SA were incubated at 37 °C under gentle shaking for various periods. SA density was obtained with a calibration curve of the optical intensity at 600 nm in a microplate spectrophotometer. For each set of assays, culture media were incubated with bacteria under the same conditions without the chips as the negative controls to normalize the bacteria density. These assays were administered from the time of seeding to data recording by fluorescence spectrometry. Assays of the inflammation-triggered efficacy of antibiotics were performed based on the optical density of the medium cultures at 600 nm. SA was also cultured in the medium with chips under inflammation-free condition to evaluate the stability of drug release.

## 3. Results and Discussion

### 3.1. Morphology of the Drug Delivery Chip

Nanoparticles are generally employed as carriers of inflammation-responsive systems of drug releases [7]. However, the issues of phagocytosis and metabolism in the human body may influence the efficiency of ROS-responsive drug delivery systems in clinical trials. The use of a subcutaneous implant chip may facilitate autonomously antimicrobial treatment without these issues. Figure 1a illustrates FESEM topographies of the silicon template with a hole array. The holes had a diameter and depth of 500 nm with a smooth surface, indicating the regularity of the designed textures. For 500H-P(MAA-DB), and a thin layer surrounding the hole edge was observed, suggesting that P(MAA-DB) grafts collapse forward into the hole edge in a dry state (Figure 1b). The hole diameter decreased from 500 to 232 nm, indicating that the length of P(MAA-DB) chains in a dry state was 134 nm. Although the holes were not capped completely, the P(MAA-DB) swelled and extended the polymer chain 2–3 times in aqueous solution to cap the holes. Thus, the length of P(MAA-DB) brush was sufficient to cross-link by BDA and form a solid cap upon the holes.

The oxygen plasma treatment facilitated the charge accumulation at the hole, resulting in higher initiator and P(MAA-DB) grafting densities. The P(MAA-DB) brush with high grafting density formed a protuberance surrounding the hole edges. After cross-linking P(MAA-DB) graft with BDA upon the holes, a thick layer upon the chip holes over a large area was observed (Figure 1c). The cross-linked P(MAA-DB) with BDA accumulated around the holes to cap the holes, as observed in the top-view FESEM image. (Figure 1c) After immersing the sample in 10 mM of H_2_O_2_ solution, the surface morphology of the chip became similar to that shown in Figure 1b, indicating the ROS-responsive behavior of the cross-linked P(MAA-DB) brush. Figure 1d,e show FESEM cross-section profiles of the 500H-P(MAA-DB)-BDA before and after immersion in the H_2_O_2_ solution. The hole chamber had a diameter and depth of 500 nm and 800 nm as drug tank, respectively. Based on the cross-sectional FESEM images, caps of P(MAA-DB)-BDA grafts was observed upon each hole but not grafted inside the holes, attributing that neither the ATRP initiator nor MAA penetrated the hole sufficiently. Because the caps of polymer network were dissociated after immersion in the H_2_O_2_ solution, the holes appear to be opened in the cross-section image of holes.

Figure 2a illustrates the 2D/3D AFM topographies and cross-section profiles of the hole array structure on silicon substrate. Morphology of the hole array was consistent with the FESEM image. For 500H- P(MAA-DB), the stretched polymer brush altered the surface textures slightly, indicating that the hole array with polymer brush did not completely match the original template. In comparison with morphology in the dry state (Figure 1b), the flower-like structure of 500H-P(MAA-DB)-BDA remained without significant changes (Figure 2b).

The hole profile did not appear in the cross-section profile because of narrowing of the holes with high aspect ratio. For 500H-P(MAA-DB)-BDA, the polymer brush surrounding the hole edge collapsed forward into the holes to cross-link with BDA as solid caps upon the holes, resulting in a complete close state of holes (Figure 2c). From the cross-section profile, the solid cap upon the holes bulged upon the holes to form the regular 500 nm polymer network, confirming that the drug could be contain inside the holes. The polymeric network of P(MAA-DB)-BDA chains could be dissociated under a low concentration of H_2_O_2_ resulting in the reopening of holes. The results suggest that the ROS-triggered valve is appropriate to apply in drug delivery systems as implant chips. Figure 3 shows the photographs of 500H, 500H-P(MAA-DB), 500H-P(MAA-DB)-BDA. Because of the regular hole array, the 500H exhibited blue color that observed along an invariable angle. (Figure 3a) For 500H-P(MAA-DB), the color turns obviously to yellow color instead of initial blue color. (Figure 3b) The results suggest that the P(MAA-DB) grafts varied the hole diameter substantially, resulting in the change of grating effect. The grating effect vanished after cross-linking P(MAA-DB) graft with BDA upon the holes, indicated the disappearance of hole array structure.

### 3.2. Surface Characterization

Appendix A (see Appendix A) illustrates the XPS survey spectra of 500H-PMAA, 500H-P(MAA-DB), and 500H-P(MAA-DB)-BDA to confirm the modification of each step. XPS spectrum of 500H-PMAA appeared peaks at the ranges of 99–104, 285–293, and 528–535 eV, attributed to Si 2p, C 1s, and O 1s peaks corresponding to the ranges of 99–104, 285–293, and 528–535 eV of the silicon wafers appeared in the XPS spectrum of PMAA, respectively. XPS spectrum of 500H-PMAA appeared peaks at the ranges of 99–104, 285–293, and 528–535 eV, attributed to Si 2p, C 1s, and O 1s peaks corresponding to the ranges of 99–104, 285–293, and 528–535 eV of the silicon wafers appeared in the XPS spectrum of PMAA, respectively. The surfaces that exhibited slight Br 3d5 peaks of the halogen group in the range of 65–73 eV could be attributed to the halogen groups of the chain end of PMAA. The surfaces of PMAA did not present N 1s peak in the range of 396–403 eV before the immobilization of DB. This peak appeared in the XPS spectrum of P(MAA-DB), verifying the presence of the DB layer [46]. A slight decrease in the carbon to oxygen intensity ratio and nitrogen elemental signal intensity was observed in the spectrum because of the coating of the DB layer. B 1s peak in the range of 190–192 eV appeared in the XPS spectrum of P(MAA-DB)-BDA, verifying the cross-linking of P(MAA-DB) with BDA.

Appendix A (see Appendix A) illustrates the FTIR spectra of PMAA, P(MAA-DB), and P(MAA-DB)-BDA. The FTIR spectrum of PMAA exhibited characteristic peaks in the range of 750–850 cm^−1^ attributed to νs(Si-O-Si); 950–1000 cm^−1^ attributed to n(Si-OH) and n(Si-O-); and 1550–1800 cm^−1^ attributed to ν(C-O), ν(Si-O-Si), and ν(Si-O-C) of silicon wafers [49]. The broad peaks of stretching vibrations of hydroxyl (COH) group of PMAA appeared in the range of 3130–3690 cm^−1^ [50]. For the FTIR spectrum of P(MAA-DB), a decrease in the intensity of broad peaks of stretching vibrations of COH group of PMAA indicated binding between carboxyl groups of PMAA and amide groups of DB. A decrease in the intensity of the carboxyl stretching vibration at 1554 cm^−1^ (secondary amide C=O stretching) can be attributed to the DB grafts on the PMAA brush. Because boronic acid derivatives can bind to diol groups, the P(MAA-DB) could be cross-linked by diboronic acid derivatives. BDA was introduced as a cross-linker to cap the holes on the silicon substrate through cross-linking the P(MAA-DB) brush with the boronate esters. Thus, adsorption wavelengths at 1272 and 1354 cm^−1^ were identified as characteristic bands of B-O to confirm the cross-linking [51].

### 3.3. Loading and Releasing of FITC-Vancomycin

FITC-vancomycin as selected as a model drug to evaluate ROS-triggered release behavior from 500H-P(MAA-DB)-BDA. The FITC-vancomycin was loaded through the hydrophilic P(MAA-DB) brush layer on the substrate into the hole array. The cross-linker, BDA, was employed to cap the hole and preserve the FITC-vancomycin inside the holes. The encapsulating efficiency (*E*) was calculated using the following equation determined using the fluorescence emission spectra:(1)$$E=\frac{V_{i}-V_{r}}{V_{h}}\times 100\%$$
where *V_i_* − *V_r_* and *V_h_* represent the initial, residual volume of FITC-vancomycin and the total space of holes of the 25-mm square chip according to our design, respectively. The value of *E* reached 97.1%, indicating the high loading efficiency of FITC-vancomycin. A 100-μL PBS solution at pH 7.4 in the presence of 10 mM H_2_O_2_ was placed on the FITC-vancomycin-loaded 500H-P(MAA-DB)-BDA surface for various periods to observe the release behavior of FITC-vancomycin.

Figure 4a–d illustrate the LSCM images of 500H-P(MAA-DB)-BDA after FITC-vancomycin release for 0, 6, 12 and 18 h. FITC-vancomycin was not extensively observed on the surface of 500H-P(MAA-DB)-BDA before ROS-triggered drug release, indicating the excellent blocking of FITC-vancomycin inside the holes (Figure 4a). After dissociating the network of P(MAA-DB)-BDA with H_2_O_2_ for 6 h, slight amounts of FITC-vancomycin were observed on the surface, indicating that FITC-vancomycin was initially released in the presence of ROS (Figure 4b). The released FITC-vancomycin continuously diffused from the hole array to the liquid phase after ROS-triggering for 12 h. The fluorescent pattern of the residual FITC-vancomycin inside the hole array that matched the hole array of the substrate was observed (Figure 4c). For releasing FTIC-vancomycin from the holes for 18 h, the high fluorescent intensity indicated the long-term drug diffusion from the hole array to the liquid phase (Figure 4d). To evaluate the stability and ROS-responsive behavior of drug release, the drug-loaded 500H-P(MAA-DB)-BDA was immersed in PBS at pH 7.4 in the absence of ROS for 4 h and H_2_O_2_ was added to the solution at various concentrations. The solutions were withdrawn at various time intervals to measure the total amount of the loaded FITC-vancomycin released based on the fluorescence intensities.

Figure 5a illustrates the cumulative release of FITC-vancomycin from the 25-mm square chip with hole array in the presence of various concentrations of H_2_O_2_ within 48 h at pH 7.4. H_2_O_2_ at various concentrations was initially added into the solution after the samples were immersed in the solution for 4 h. Only 1.4% FITC-vancomycin was released without ROS-triggering within the first 4 h, indicating the excellent blocking performance of the BDA-cross-linked 500H-P(MAA-DB)-BDA system. The released FITC-vancomycin in the solution reached merely 6.4% after 48 h in the absence of H_2_O_2_. The release of FITC-vancomycin was abruptly increased at concentrations of 5, 10, 15 and 20 mM owing to the network dissociation of P(MAA-DB)-BDA. The release of FITC-vancomycin in the presence of 15 and 20 mM of H_2_O_2_ increased abruptly in real-time from 4 to 30 h, gradually reaching a plateau from 30 to 48 h. The release rate of FITC-vancomycin with 10 mM of H_2_O_2_ was markedly reduced. After ROS-triggering, 74.8% of the drug was released gradually within 48 h, indicating that H_2_O_2_ concentration is markedly associated with the release rate of FITC-vancomycin. We observed a linear increase in FITC-vancomycin release from 0 to 38.9% at 5 mM H_2_O_2_ solution in real time from 4 to 48 h, implying the long-term drug release. In addition, the coordination between the functional groups of boronic acid and carboxyl groups on the PMAA chain also decreased the pKa, which may be influenced by pH values [52]. Figure 5b illustrates the cumulative release of FITC-vancomycin from the 25-mm square chip with hole array at various pH values for 48 h. Only 1.4% and 2.1% FITC-vancomycin were released from the chip after 48 h of immersion in solutions of pH 8.1 and 7.4, respectively. The release of FITC-vancomycin reached 31.2% and 59.11% after 48 h of immersion in solutions of pH 6.1 and 5.2, respectively (Figure 5b). These results suggest that FITC-vancomycin was also released from the chip in an acidic environment, consistent with the findings of previous studies [46,53].

The compatibility of an implant chip is an important factor for facilitating host cell adhesion, spreading, and permanent growth. Heparin L929 cells were seeded on the tissue culture plate and sample surfaces to examine the cell affinity with cell proliferation by CCK-8 assay at 1, 3, 5 and 7 days. Optical density at 600 nm after cell proliferation on the tissue culture plate at various periods was employed as a standard to calculate the cell viability ratio of sample surfaces.

Figure 6a shows cell viability ratio proliferated on the sample surface measured by CCK-8 assay within 7 days. The cell viability ratio of 500H surface ranged from 34.6 to 42.3% indicating poor ability of cell proliferation. The ability of cell proliferation, ranged from 82.3 to 86.6%, was enhanced significantly with the PMAA graft. Additionally, the cell viability ratio of 500H-P(MAA-DB) surface exhibited a tendency of increase, which indicated that DB molecules possessed better compatibility than PMAA. After cross-linking 500H-P(MAA-DB) with BDA, the cell viability ratio of the surface slightly decreased, which might be ascribed to the change in morphology. All the chips can support cell adhesion and proliferation without toxicity. To evaluate the reversibility of the polymer-capped system, 500H-P(MAA-DB)-BDA, after the cleavage of the phenylboric acid ester bond following treatment with 10 mM of H_2_O_2_ for 48 h, was loaded the drug reversibly, reacted with BDA to cap the hole, and reused with immersion in 10 mM of H_2_O_2_ for 48 h. Figure 6b illustrates the amount of FITC-vancomycin that can be loaded inside the chip and released from the chip in five cycles. The ability of the chip to store FITC-vancomycin decreased slightly from 43.7–52.4 μg/cm^2^ after the drug was reloaded reversibly for five cycles. FITC-vancomycin concentration inside the chip after ROS-triggered release for 48 h remained in the range of 3.3–4.5 μg/cm^2^. The excellent reversibility of FITC-vancomycin loading and release through the polymer-capped system suggests that the stable hole opening and closing behavior can be switched to release and block FITC-vancomycin reversibly using H_2_O_2_ and BDA, respectively. P(MAA-DB) cross-linked with BDA can satisfactorily gate the hole array with strong potential for application in antimicrobial treatment.

LPS was employed to activate macrophages, resulting in both pH and ROS stimuli that can simulate inflammatory microenvironment in vitro. FITC-vancomycin was encapsulated into the 500H-P(MAA-DB) with seal of polymeric network as a model drug to investigate the behavior of inflammation-responsive drug release. Fluorescence of the drug did not be observed with seal of polymer network; however, it could be observed after release from the hole array and internalization in the cellular esterase. Macrophages with LPS activation were incubated on the drug-loaded chips to evaluate the performance of the drug release. In addition, the LPS-free macrophages with the chip in the media were considered as a control.

Figure 7a,b shows the fluorescence microscopy images of in LPS-activated and LPS-free macrophages with the 500H-P(MAA-DB)-BDA, respectively. The macrophages without LPS activation did not exhibit significant fluorescence in the image. The obvious fluorescence of macrophages with LPS activation appeared in the image, indicating the excellent performance of inflammation-responsive drug release. Figure 7c shows that the normalized fluorescence of LPS-activated and LPS-free macrophages plotted as a function of time on the chips. The fluorescence intensity of macrophages with LPS activation increased gradually from 0 to 86.9% with time from 0 to 2 h and reached a plateau from 2 to 3 h, indicating that 2 h of the optimal incubation time of the inflammation-triggered drug release. While the fluorescence intensity of LPS-free macrophages did not increase significantly, suggested that the FITC-vancomycin was contained inside the hole array stably. The fluorescence intensity of macrophages with LPS activation for 4 h was 9.1 times higher than that without LPS activation, suggested that the pH/ROS dual-responsive chip release drug in an inflammatory environment. Furthermore, we evaluated inflammation-triggered efficacy of antibiotics released from the chips and exhibited the potential of the chip as drug delivery vehicles. SA, a common bacterium of biomedical implant infections, was employed in vitro assessments of the therapeutic activities. The drug-loaded 500H-P(MAA-DB)-BDA chip were incubated with LPS-activated and LPS-free macrophages for 2 h, and then coincubated with SA sample. Figure 7d shows the dependence of normalized bacteria density on the culture time. Normalized bacteria density (*NBD*) was calculated as follows:(2)$$NBD=\frac{OD_{chip}-OD_{blank}}{OD_{control}-OD_{blank}}$$
where *OD_chip_* and *OD_control_* represented the optical density of SA that cultured with and without the chips at 600 nm, respectively. While *OD_blank_* represented the optical density of blank medium. The SA density in the medium with the chip remained 100% within 6 h under inflammation-free condition, indicating that chip did not release antibiotics significantly under inflammation-free condition to suppress the SA growth. The SA density decreased markedly in the medium with the inflammation-triggered chip, suggesting that SA growth was suppressed obviously due to release of the antibiotic from the chip. These results verified the behavior of inflammation-triggered antibiotic release to inhibit SA growth for 500H-P(MAA-DB)-BDA chip. 

## 4. Conclusions

PMAA brushes were grafted from a 500-nm hole array on a silicon wafer via SI-ATRP to immobilize with DB through an EDC/NHS reaction. The P(MAA-DB)-modified hole array loaded with FITC-vancomycin inside was capped by cross-linking between BDA and P(MAA-DB) to block the release of FITC-vancomycin. The polymer brush caps upon the 500-nm holes of the substrate were cleaved under the stimuli of ROS to release FITC-vancomycin from the holes via diffusion. The dissociation of the polymer brush network was rebound by BDA reversibly to cap the hole for recycling. The reversible hole closing and opening with BDA and ROS could block and release the drug, respectively, which are appropriate for applications in antibiotic storage and delivery systems. The chips were also exhibited the excellent in vitro efficacy against bacteria growth without significant toxicity toward the cells. Our proposed chip could be more convenient than antibiotic-loaded nanoparticles for applications in clinical trials without phagocytosis- and metabolism-related issues. Although the amount of antibiotic loaded inside the hole array is not remarkable, the space of each hole can be extended using dry etching technology to enhance the storage capacity. The proposed antibiotic storage and delivery system implant chip not only exhibits smart polymer valves but also excellent biocompatibility, recyclability, and stability, which can be easily extended to other drug deliver system for numerous biomedical applications.

## Data Availability

The authors declare that data related to this study are provided upon request.

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
