# Peer review of "Inflammation-Responsive Nanovalves of Polymer-Conjugated Dextran on a Hole Array of Silicon Substrate for Controlled Antibiotic Release"

_polymers, 2022, doi:10.3390/polym14173611_

Round 1

Reviewer 1 Report

The manuscript "Inflammation-responsive nanovalve of polymer-conjugated dextran on hole array of silicon substrate for controlled antibiotic release" by J.K. Chen et al. reports on the preparation of dextran-biotin conjugates immobilized on the PMAA chains as caps covering the hole edges on the silicon surface embedding antibiotics inside the holes. The idea of the paper is very good and the experimental data are well described supporting the results and conclusions.

There are some minor issues regarding the manuscript:

- why authors selected FITC labeled vancomycin as model of antibiotic for these materials?

-why did authors select pH 5.2, 6.1, 7.4 and 8.1 to evaluate the pH-responsive behavior of the drug release?

-for loading capacity which was the concentration of FITC-vancomycin stock solution? This data should be added in the Section Loading and releasing of FITC-vancomycin.

-how authors did correlate the release data with the compatibility of the implant, knowing that the cell culture L929 viability was studied at pH 7.4?

-Did the released concentration of FITC labeled vancomycin is sufficient to promote its antibacterial activity?

-Did the proposed chip is advantageous over other materials used in biomedical applications, regarding the preparation and marketing costs?

On the basis of these observations I recommend the publication of this manuscript after Minor revision.

Author Response

Reviewer's comments:

The manuscript "Inflammation-responsive nanovalve of polymer-conjugated dextran on hole array of silicon substrate for controlled antibiotic release" by J.K. Chen et al. reports on the preparation of dextran-biotin conjugates immobilized on the PMAA chains as caps covering the hole edges on the silicon surface embedding antibiotics inside the holes. The idea of the paper is very good and the experimental data are well described supporting the results and conclusions.

There are some minor issues regarding the manuscript:

- why authors selected FITC labeled vancomycin as model of antibiotic for these materials?

Our reply: FITC labeled vancomycin is a commercial antibiotic that can be used to monitor the trace of drug release.

-why did authors select pH 5.2, 6.1, 7.4 and 8.1 to evaluate the pH-responsive behavior of the drug release?

Our reply: The pH values range from 6 to 7 and 5.6 to 6.8 generally in the inflammatory environment and microenvironment of tumors, respectively. To investigate the pH-responsive behavior of drug release in more broad range, we prepared environments at pH 5.2, 6.1, 7.4 and 8.1. Buy the way, the PBS is a kind of buffer solution, pH values are difficult to control precisely at integers.

-for loading capacity which was the concentration of FITC-vancomycin stock solution? This data should be added in the Section Loading and releasing of FITC-vancomycin.

Our reply: Pure FITC-vancomycin was employed to load within the 25-mm square chip. The loading capacity was calculated by gravimetric analysis. Loading capacity represents how much FITC-vancomycin could be loaded within the chip. We have provided the data in the Section Loading and releasing of FITC-vancomycin.

-how authors did correlate the release data with the compatibility of the implant, knowing that the cell culture L929 viability was studied at pH 7.4?

Our reply: The experimental purpose of cell culture L929 viability at pH 7.4 was evaluate the cytotoxicity of the chips to confirm the enhancement in compatibility after the modification. We did not correlate the release data with the compatibility of the implant. We have revised the typo sentence "Drug release was biocompatible…" as "P(MAA-DB) brush-modified chip was biocompatible…" in the abstract. 

-Did the released concentration of FITC labeled vancomycin is sufficient to promote its antibacterial activity?

Our reply: The loading capacity of the chip is 52.4 mg/cm2. The antibiotic concentration is ca. 20-50 mg/g per day that is sufficient to promote its antibacterial activity. The chip seems insufficient slightly to promote its antibacterial activity in vivo. We are enlarging the hole size as well as loading capacity of the chip recently to achieve the sufficient dosage and test the antibacterial activity in vivo.

-Did the proposed chip is advantageous over other materials used in biomedical applications, regarding the preparation and marketing costs?

Our reply: Most of other materials are fabricated as particles to encapsulate the antibiotic. Our proposed chip could be more convenient than antibiotic-loaded nanoparticles for applications in clinical trials without phagocytosis- and metabolism-related issues, which may provide a new concept to develop a drug carrier instead of particles. It may take 20 years to develop a drug to a commercial product. Preparation and marketing costs should not be the main reasons to determine development of the drug. Fabrication processes of chips are well-developed now, indicating that preparation and marketing costs of chips in mass production should be lower than that of particles.

Reviewer 2 Report

Lee et al. they present an interesting and well-structured manuscript. The introduction to the state of the art is correct and with adequate references, I would suggest that the authors give more importance to the innovation and novelty that this manuscript brings. The methodology is clear and reproducible.

The results section is correct and is presented in an understandable way.

The suggestion is that this manuscript has enough substance to have its own independent discussion.

I would suggest the authors extensively review the English grammar

Author Response

Reviewer's comments:

Lee et al. they present an interesting and well-structured manuscript. The introduction to the state of the art is correct and with adequate references, I would suggest that the authors give more importance to the innovation and novelty that this manuscript brings. The methodology is clear and reproducible.

The results section is correct and is presented in an understandable way.

The suggestion is that this manuscript has enough substance to have its own independent discussion.

I would suggest the authors extensively review the English grammar

Our reply: The manuscript has been edited by a native English speaker.